# Uncovering the Building Blocks of ECGs with a Discrete Autoencoder

## Abstract

Automated electrocardiogram (ECG) interpretation is crucial for cardiovascular diagnostics, with ECG self-supervised learning (eSSL) emerging as a powerful paradigm to exploit large-scale unlabeled datasets. However, current eSSL frameworks suffer from two key limitations: their learned representations are typically continuous, high-dimensional and opaque, hindering clinical trust; and they are anchored to coarse, human-defined waveform concepts, thus limiting the model's intrinsic capacity to discover novel biomarkers from finer sub-waveform morphologies. To address these issues, we propose **AtomECG**, an eSSL framework that views the ECG as a discrete sequence of fundamental and reusable morphological **atoms** governed by an underlying **grammar**. Specifically, we introduce **Two-Scale Manifold Alignment**, a novel quantization scheme. By simultaneously learning a "grammar manifold" for the entire codebook and employing a geometry-aware alignment for individual patches, AtomECG maps finer sub-waveform morphologies to discrete atoms. Extensive experiments demonstrate that AtomECG not only achieves state-of-the-art performance on a wide range of diagnostic tasks but also provides strong interpretability by explicitly mapping specific atoms to pathological patterns. Furthermore, AtomECG shows potential for long-term monitoring and demonstrates robust generalization across diverse patient populations, underscoring its promise for clinical deployment.

## 1 Introduction

The electrocardiogram (ECG) is an indispensable diagnostic tool, offering a non-invasive window into the heart's electrical activity (Goldberger, 2018). However, the manual interpretation of ECGs is a complex skill demanding extensive training, while the development of automated systems is severely bottlenecked by the scarcity of large-scale, high-quality annotated data (Schläpfer & Wellens, 2017). To bridge this gap, ECG self-supervised learning (eSSL) has emerged as a compelling paradigm (Mehari & Strodthoff, 2022). By pre-training on vast amounts of unlabeled data, these models can accumulate foundational knowledge of cardiac patterns, much like human experts build experience, before being fine-tuned for specific downstream tasks.

However, despite their empirical success, current eSSL frameworks face two fundamental limitations. First, most methods produce continuous, high-dimensional representations, limiting interpretability in clinical contexts where understanding the rationale is as vital as the diagnosis. Second, existing methods, whether implicitly or explicitly, often operate at a coarse granularity dictated by human-defined fiducial points (e.g., P-wave, QRS complex, and T-wave). This reliance on macroscopic waveform risks discarding potentially diagnostic information embedded in subtle, sub-waveform morphologies that fall outside conventional clinical definitions. Ideally, models should move beyond these constraints and be able to identify novel, data-driven biomarkers from the signal's most fundamental building blocks.

To address these challenges, we posit that a complex ECG waveform can be viewed not as a continuous signal, but as a sequence composed of fundamental, reusable morphological primitives, which we term **ECG atoms**. These atoms, governed by an underlying **ECG grammar**, combine to form the intricate patterns observed in clinical recordings. From this atomic view, pathological patterns emerge as changes in atoms or their composition, with diagnostic insights traceable to specific atoms

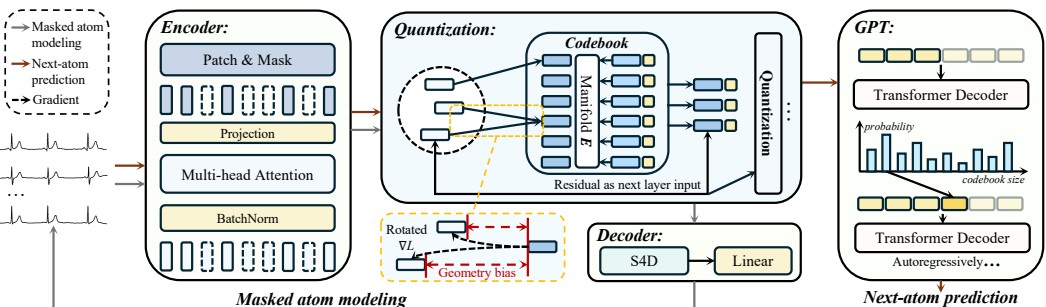

Figure 1: The framework of AtomECG.

or grammar rules. Furthermore, by defining atoms at a sub-beat level, the model can discover fundamental units directly from data, unconstrained by human-defined boundaries.

To realize this vision, we propose **AtomECG**, an eSSL framework that learns a discrete vocabulary of ECG atoms. As shown in Figure 1, AtomECG begins by partitioning the raw signals into uniform patches, with a length chosen to be significantly smaller than a typical cardiac cycle to capture atomic morphologies. We employ a learnable codebook and quantize each patch to an atom. To overcome the ambiguity between local patch similarity and global cardiac context across different cardiac cycles, we propose a novel quantization scheme, **Two-Scale Manifold Alignment**. By learning a globally coherent "grammar manifold" for the codebook while using a local, geometry-aware alignment to ensure contextually precise mapping for each patch, this mechanism learns finer-grained atom representations and a structured atom space, enabling a more faithful expression of ECG grammar. Following BERT (Devlin et al., 2019), AtomECG employs masked atom modeling for pre-training, aiming to obtain generalizable representations.

We conduct extensive experiments to validate the effectiveness of AtomECG. First, across more than ten classification tasks, AtomECG outperforms eight leading eSSL baselines, establishing a new SOTA. Second, AtomECG uncovers disease-specific atoms and sequences that closely correspond to known pathological patterns, demonstrating strong interpretability. Finally, we verify the ECG grammar through next-atom prediction and assess AtomECG's robustness in long-term monitoring and cross-population generalization, where it demonstrates strong and reliable performance. Our primary contributions are summarized in below:

- We propose a novel atomic perspective and introduce AtomECG, a simple eSSL framework that learns a discrete vocabulary of morphological atoms and their underlying grammar.
- We introduce Two-Scale Manifold Alignment, a novel quantization mechanism that enables AtomECG to achieve SOTA diagnostic performance.
- We demonstrate that the learned atoms and their sequences offer high clinical interpretability and are governed by a learnable grammar.
- We validate the model's practical utility on long-term monitoring and cross-population generalization, highlighting its potential for real-world deployment.

## 2 RELATED WORK

**eSSL**. eSSL methods can be primarily categorized into contrastive and generative approaches. For contrastive learning, the classic framework MoCo (He et al., 2020) and SwAV (Caron et al., 2020) have been successfully adapted for ECG pre-training. Kiyasseh et al. (2021) redefines positive pairs to capture patient-level shared context beyond traditional instance-level context, proposing three key modes: CMSC (adjacent temporal segments as positive pairs), CMLC (different leads at the same time point as positive pairs), and CMSMLC (combining temporal and spatial invariance). Other contrastive methods mainly differ in their strategies for constructing positive/negative pairs and data augmentation. For instance, Wang et al. (2024) designs a novel noise-based augmentation scheme and an adversarial contrastive strategy. Lan et al. (2022) incorporates physiological

differences by performing contrastive learning both within and across individuals. For generative learning, Zhang et al. (2022) introduces three masking strategies: MTAE (masking along the time dimension), MLAE (masking along the lead dimension), and MLTAE (spatio-temporal masking). RLM (Oh et al., 2022) is another random lead masking strategy that simulates real-world scenarios of incomplete ECG leads, enabling the model to adapt to arbitrary lead combinations. ST-MEM (Na et al., 2024) decomposes the ECG into patches considering both lead and time dimensions, introduces lead embeddings and a [SEP] token, and finally decodes each lead separately. Furthermore, some works incorporate auxiliary information to aid downstream diagnosis (Jiang et al., 2024). Liu et al. (2024) leverages clinical knowledge from ECG reports, using the large language model to assist in diagnosis.

**ELP**(ECG Language Processing). Contrastive methods that rely on augmentations and positive-negative discrimination primarily learn continuous, high-dimensional representations, which hinders their interpretability. To address this limitation, some approaches split ECGs into sequences of patches, leveraging NLP techniques to yields more interpretable representations for downstream tasks. However, early work such as ELP (Mousavi et al., 2021) does not employ SSL to construct a comprehensive vocabulary, thus missing more general ECG representations. Subsequent generative eSSL methods like ECGBERT (Choi et al., 2023) and HeartLang (Jin et al., 2025) achieve a degree of interpretability through waveform segmentation and masked ECG language modeling. Nevertheless, their operational granularity is constrained by the human-defined fiducial points. This reliance on macroscopic landmarks risks discarding potentially crucial diagnostic information embedded in sub-waveform morphologies that lie outside of conventional clinical definitions.

**Quantization**. As the core computation in AtomECG, quantization shares the same objective as VQ-VAE (van den Oord et al., 2017): to obtain high-quality discrete latent representations. Improving quantization accuracy and codebook utilization are two critical optimization directions. To enhance quantization accuracy, SoundStream (Zeghidour et al., 2021) recursively quantizes the waveform's residual, while HiFi-Codec (Yang et al., 2023) performs grouped quantization on the features. To improve codebook utilization, SimVQ (Zhu et al., 2024) employs a linear transformation of the codebook to effectively propagate updates. The Rotation Trick (Fifty et al., 2024) mitigates the risk of codebook collapse by optimizing gradient directions. FSQ (Mentzer et al., 2023) simplifies the quantization process, where the codebook vectors become uniform points within a hypercube.

# 3 METHOD

In this section, we first describe the design principle of the encoder and decoder, and then introduce how the proposed quantization method, Two-Scale Manifold Alignment, constructs a meaningful codebook of ECG atoms at both global and local levels.

## 3.1 GENERATIVE FRAMEWORK

**Patch and Mask**. Given an ECG signal $\mathbf{x} \in \mathbb{R}^{c \times l}$ with patch length $b$ and stride $s$, the number of resulting patches is $p = \lfloor \frac{l-b}{s} \rfloor + 1$. Under the pre-training task of masked atom modeling, we randomly mask $k \times p$ patches. To ensure a fair comparison, we fix the total amount of corrupted information in the signal rather than merely controlling the number of masked patches.

**Encoder and Decoder**. AtomECG employs an asymmetric encoder-decoder pre-training framework. The encoder is guided to focus on capturing high-dimensional information crucial for complex downstream diagnostics, such as signal periodicity and intricate patterns. Conversely, the decoder is tasked with processing simpler, low-dimensional information, like waveform amplitude. By assigning the decoder to reconstruct fundamental signal attributes, the encoder is incentivized to learn a discriminative and global ECG representation.

We employ the Multi-Head Attention (Vaswani et al., 2017) as the encoder. The advantage lies in projecting the input sequence into multiple, parallel representation subspaces and computing attention weights independently within each subspace. This parallel processing enables the model to simultaneously capture diverse types of dependencies within the ECG signal (e.g., short-term morphological features and long-term rhythmic patterns) and to fuse these multifaceted perspectives into

---

**Algorithm 1** Pre-training Procedure of AtomECG.

---

**Require**: Input signal $\mathbf{x} \in \mathbb{R}^{l \times c}$.
**Require**: $h$ separate codebooks $\{\mathcal{C}^{(i)}\}_{i=0}^{h-1}$, where $\mathcal{C}^{(i)} = \{\mathbf{c}_1, \ldots, \mathbf{c}_N\}$, $\mathbf{c}_n \in \mathbb{R}^d$.
**Require**: $h$ separate linear transformations $\{\mathbf{E}^{(i)}\}_{i=0}^{h-1}$, where $\mathbf{E}^{(i)} \in \mathbb{R}^{d \times d}$.
  1: $\mathbf{f} = \text{Encoder}(\mathbf{x})$, $\mathbf{f} \in \mathbb{R}^{M \times d}$, $M = c \times p$
  2: $\mathbf{r}^{(0)} = \mathbf{f}$, $\hat{\mathbf{f}}_{\text{total}} = \mathbf{0} \in \mathbb{R}^{M \times d}$
  3: **for** $i \leftarrow 0$ **to** $h - 1$
  4:     $\mathbf{Q}^{(i)} \in \mathbb{R}^{M \times d}$, $\hat{\mathbf{f}}^{(i)} \in \mathbb{R}^{M \times d}$
  5:     **for** $j \leftarrow 0$ **to** $M - 1$
  6:         $n_j^* = \underset{n=1,\ldots,N}{argmin} \|\mathbf{r}_j^{(i)} - \mathbf{c}_n^{(i)}\mathbf{E}^{(i)}\|$
  7:         $\mathbf{Q}_j^{(i)} = \mathbf{q}_j$, $\mathbf{q}_j = \mathbf{c}_{n_j^*}^{(i)}$ {Select the best atom and store it for residual computation.}
  8:         $\mathbf{R}_{\theta,j} = \frac{\|\mathbf{q}_j\mathbf{E}^{(i)}\|}{\|\mathbf{r}_j^{(i)}\|}(\mathbf{I} - 2\frac{\mathbf{w}\mathbf{w}^T}{\mathbf{w}^T\mathbf{w}+\epsilon})$, $\{\mathbf{w} = \mathbf{u} - \mathbf{v}, \mathbf{u} = \frac{\mathbf{q}_j\mathbf{E}^{(i)}}{\|\mathbf{q}_j\mathbf{E}^{(i)}\|}, \mathbf{v} = \frac{\mathbf{r}_j^{(i)}}{\|\mathbf{r}_j^{(i)}\|}\}$
  9:         $\hat{\mathbf{f}}_j^{(i)} = \text{sg}[\mathbf{R}_{\theta,j}]\mathbf{r}_j^{(i)}$ {sg(stop gradient).}
  10:     **end for**
  11:     $\mathbf{r}^{(i+1)} = \mathbf{r}^{(i)} - \mathbf{Q}^{(i)}\mathbf{E}^{(i)}$
  12:     $\hat{\mathbf{f}}_{\text{total}} = \hat{\mathbf{f}}_{\text{total}} + \hat{\mathbf{f}}^{(i)}$
  13: **end for**
  14: $\hat{\mathbf{x}}^{c \times l} = \text{Decoder}(\underset{p \times d \rightarrow l}{\text{Linear}}(\hat{\mathbf{f}}_{\text{total}}))$
  15: **Return** $\hat{\mathbf{x}}$

---

a comprehensive, high-dimensional feature vector. And the simplified calculation can be written as:

$$\mathbf{f}^{d \times p \times c} = \text{Multihead}(\underset{b \rightarrow d}{\text{Linear}}(\mathbf{x}^{b \times p \times c})), \text{where head}_i = \text{Softmax}\left(\frac{(\mathbf{x}\mathbf{W}_i^Q)(\mathbf{x}\mathbf{W}_i^K)^T}{\sqrt{c_h}}\right)(\mathbf{x}\mathbf{W}_i^V).$$

Accurately modeling low-level ECG information requires addressing the challenges of multi-periodicity and long sequences. Alcaraz & Strodthoff (2022) demonstrates that S4(Gu et al., 2021) achieves SOTA performance on various long-range ECG imputation and forecasting tasks. Furthermore, given the efficient long-range dependency modeling capabilities of State Space Models (SSM) and their advantages over RNNs and Transformers in maintaining historical context, we adopt S4D(Gu et al., 2022) as the primary component of our decoder. Algorithm details are shown in Appendix A.1.1.

### 3.2 QUANTIZATION VIA TWO-SCALE MANIFOLD ALIGNMENT

Once the encoder maps the input signal $\mathbf{x}$ to a sequence of patch representations $\mathbf{f}$, the next crucial step is to assign each patch representation to a discrete atom from a codebook $\mathcal{C}$. For periodic signals like ECG, this process faces a fundamental challenge rooted in the ambiguity between local similarity and global context. Specifically, we identify three critical scenarios: (a)Healthy, periodic patterns where similar patches across different cardiac cycles should map to the same atom. (b)Pathological patterns that are consistent across cycles (e.g., a persistent bundle branch block), where the abnormal patches should map to a consistent, disease-specific atom. (c)Irregular pathological events (e.g., a premature ventricular contraction), where a patch, despite its local similarity to a healthy counterpart, must be mapped to a distinct atom due to its anomalous context.

Standard nearest-neighbor quantization, which minimizes the Euclidean distance $\|\mathbf{f}_j - \mathbf{c}_n\|^2$, operates solely on local similarity. It is inherently ill-equipped to distinguish between these scenarios, often mapping contextually different patches to the same atom, thereby losing critical diagnostic information. To resolve this ambiguity, we introduce a novel quantization scheme named **Two-Scale Manifold Alignment** as shown in Algorithm 1, which operates on both a global and a local level to learn a truly meaningful vocabulary of ECG atoms.

**Global Scale: Learning a Shared Grammar Manifold**. At the global scale, our goal is to impose a structural prior on the codebook, ensuring that the learned atoms collectively form a coherent

representation space that reflects the underlying "grammar" of ECG signals. Instead of using a static codebook $\mathcal{C}$, we introduce a learnable linear transformation $\mathbf{E} \in \mathbb{R}^{d \times d}$. The matching process is then performed in a dynamically transformed space, finding the atom $\mathbf{c}_{n_j^*}$ that minimizes the distance to the patch representation $\mathbf{r}_j$: $n_j^* = \operatorname{argmin}_{n=1,\ldots,N} \|\mathbf{r}_j - \mathbf{c}_n \mathbf{E}\|_2^2$. The matrix $\mathbf{E}$ transforms the entire discrete codebook $\mathcal{C}$ into a dynamic "grammar manifold", defined as $\mathcal{M}_\mathcal{C} = \{\mathbf{cE} \mid \mathbf{c} \in \mathcal{C}\}$. This manifold is optimized based on the collective statistics of the data. The gradient with respect to $\mathbf{E}$ aggregates information from all selected atoms within a batch:

$$\nabla_{\mathbf{E}} L = \sum_{j=0}^{M-1} \nabla_{\mathbf{E}} L(\mathbf{r}_j, \mathbf{c}_{n_j^*} \mathbf{E}). \tag{1}$$

This collective update mechanism ensures that all atoms, even those not selected in a given step (as their effective positions $\mathbf{c}_n \mathbf{E}$ are shifted), co-evolve in a structured manner. This process sculpts the manifold $\mathcal{M}_\mathcal{C}$ to optimally represent the global distribution of valid ECG morphologies, enhancing atom utilization and preventing codebook collapse.

**Local Scale: Context-Aware Geometric Alignment**. While $\mathbf{E}$ establishes the global structure, we need a local mechanism to perform context-aware matching for each individual patch. This is crucial for distinguishing patches that are mapped to the same atom but exhibit subtle differences. Simple gradient-passing techniques like the Straight-Through Estimator(STE) are context-agnostic, providing a uniform "pull-closer" signal regardless of the geometric relationship between a patch and its target atom. We apply a more informative gradient steering mechanism based on the Householder transformation (Golub & Van Loan, 2013). For each $\mathbf{r}_j$ and its matched atom $\mathbf{q}_j = \mathbf{c}_{n_j^*}$, we construct a transformation matrix $\mathbf{R}_{\theta,j}$ that aligns $\mathbf{r}_j$ with its target $\mathbf{q}_j \mathbf{E}$ in both direction and magnitude. Let $\mathbf{u} = \frac{\mathbf{q}_j \mathbf{E}}{\|\mathbf{q}_j \mathbf{E}\|}$ and $\mathbf{v} = \frac{\mathbf{r}_j}{\|\mathbf{r}_j\|}$ be the normalized vectors. The transformation is:

$$\mathbf{R}_{\theta,j} = \frac{\|\mathbf{q}_j \mathbf{E}\|}{\|\mathbf{r}_j\|} \left( \mathbf{I} - 2 \frac{(\mathbf{u} - \mathbf{v})(\mathbf{u} - \mathbf{v})^T}{(\mathbf{u} - \mathbf{v})^T(\mathbf{u} - \mathbf{v}) + \epsilon} \right). \tag{2}$$

The term in the parenthesis is a Householder matrix that reflects $\mathbf{v}$ onto $\mathbf{u}$. During backpropagation, the gradient flows from the quantized representation $\hat{\mathbf{f}}_j = \operatorname{sg}[\mathbf{R}_{\theta,j}]\mathbf{r}_j$ to the encoder output $\mathbf{r}_j$ as:

$$\nabla_{\mathbf{r}_j} L = \mathbf{R}_{\theta,j}^T \nabla_{\hat{\mathbf{f}}_j} L. \tag{3}$$

Unlike the identity matrix used in STE, $\mathbf{R}_{\theta,j}$ is an input-dependent operator. The resulting gradient is geometry-aware: it not only pulls $\mathbf{r}_j$ towards its target but also actively rotates it, with the rotational component being larger when the angular deviation between $\mathbf{r}_j$ and $\mathbf{q}_j \mathbf{E}$ is greater. This sensitivity to angular discrepancy is precisely what allows the encoder to learn representations that distinguish subtly different but diagnostically critical morphologies.

In synergy, the global grammar manifold learned by $\mathbf{E}$ provides a stable, structured space of possible atoms, while the local geometric alignment via $\mathbf{R}_{\theta,j}$ ensures that the mapping of each patch to this space is contextually precise. This two-scale approach resolves the local-global ambiguity, enabling the model to learn a robust and interpretable set of building blocks from complex periodic signals. Additionally, we introduce residual quantization to enhance accuracy. The quantization error from one layer is passed to the next for further quantization. The quantization details are shown in Appendix A.1.2.

**Loss.** We employ MSE as the primary loss for masked atom modeling, while simultaneously training the codebook with the vq and commitment losses from VQ-VAE (van den Oord et al., 2017). So the loss function in pre-training can be written as $\mathcal{L} = \operatorname{MSE}(\mathbf{x}, \hat{\mathbf{x}}) + \mathcal{L}_{\text{vq}} + \mathcal{L}_{\text{commit}}$, where

$$\mathcal{L}_{\text{vq}} = \sum_{i=0}^{h-1} \left( \alpha_i \|\operatorname{sg}(\mathbf{r}^{(i)}) - \mathbf{Q}^{(i)} \mathbf{E}^{(i)}\|_2^2 \right), \quad \mathcal{L}_{\text{commit}} = \sum_{i=0}^{h-1} \left( \beta_i \|\mathbf{r}^{(i)} - \operatorname{sg}(\mathbf{Q}^{(i)} \mathbf{E}^{(i)})\|_2^2 \right). \tag{4}$$

## 4 EXPERIMENTS

### 4.1 DATASETS AND EXPERIMENTAL SETTINGS

We utilize four datasets from PhysioNet(Goldberger et al., 2000) and VitalDB(Lee et al., 2022). All experiments were conducted on a NVIDIA H200 GPU equipped with 141GB memory. In pre-

Table 1: Macro AUROC of AtomECG and eight eSSL models via linear probing across seven multi-label tasks on PTB-XL and G12EC.

| Model | all | diag. | sub. | super. | form | rhythm | G12EC |
|---|---|---|---|---|---|---|---|
| *contrastive SSL* | | | | | | | |
| MoCo v3 | 0.704 | 0.750 | 0.763 | 0.777 | 0.716 | 0.734 | 0.929 |
| SwAV | 0.712 | 0.744 | 0.753 | 0.754 | 0.730 | 0.729 | 0.931 |
| CMSC | 0.684 | 0.732 | 0.742 | 0.760 | 0.723 | 0.721 | 0.930 |
| CMSMLC | 0.679 | 0.721 | 0.751 | 0.760 | 0.727 | 0.758 | 0.931 |
| *generative SSL* | | | | | | | |
| MTAE | 0.599 | 0.705 | 0.623 | 0.652 | 0.603 | 0.671 | 0.849 |
| MTAE+RLM | 0.536 | 0.688 | 0.621 | 0.614 | 0.472 | 0.579 | 0.805 |
| ST-MEM | 0.629 | 0.702 | 0.643 | 0.704 | 0.658 | 0.756 | 0.886 |
| HeartLang | 0.796 | 0.821 | 0.861 | 0.853 | 0.786 | **0.884** | 0.926 |
| **AtomECG** | **0.812** | **0.825** | **0.869** | **0.886** | **0.807** | 0.882 | **0.950** |

Table 2: Macro F1 of AtomECG and nine other models under nonlinear probing across four multi-class tasks on PTB-XL and G12EC. [1]Atrial Fibrillation, [2]First Degree Atrioventricular Block, [3]Left Bundle Branch Block, [4]Right Bundle Branch Block.

| Model | AF[1] | I-AVB[2] | LBBB[3] | RBBB[4] | AF[1] | I-AVB[2] | LBBB[3] | RBBB[4] |
|---|---|---|---|---|---|---|---|---|
| | *Diagnosis on PTB-XL* | | | | *Diagnosis on G12EC* | | | |
| *supervised baseline* | | | | | | | | |
| ResNet-18 | 0.862 | 0.620 | 0.874 | 0.766 | 0.904 | 0.909 | 0.833 | 0.832 |
| *contrastive SSL* | | | | | | | | |
| MoCo v3 | 0.920 | 0.919 | 0.932 | 0.929 | 0.923 | 0.920 | 0.930 | 0.929 |
| SwAV | 0.940 | 0.923 | 0.931 | 0.938 | 0.949 | 0.925 | 0.931 | 0.950 |
| CMSC | 0.906 | 0.908 | 0.910 | 0.910 | 0.910 | 0.911 | 0.920 | 0.919 |
| CMSMLC | 0.937 | 0.935 | 0.942 | 0.942 | 0.947 | 0.945 | 0.950 | 0.948 |
| *generative SSL* | | | | | | | | |
| MTAE | 0.919 | 0.910 | 0.921 | 0.923 | 0.924 | 0.924 | 0.920 | 0.926 |
| MTAE+RLM | 0.928 | 0.926 | 0.935 | 0.936 | 0.930 | 0.931 | 0.935 | 0.934 |
| ST-MEM | 0.925 | 0.924 | 0.930 | 0.931 | 0.928 | 0.926 | 0.934 | 0.935 |
| HeartLang | 0.935 | 0.934 | 0.936 | 0.934 | 0.940 | 0.944 | 0.945 | 0.947 |
| **AtomECG** | **0.951** | **0.952** | **0.965** | **0.966** | **0.953** | **0.955** | **0.964** | **0.964** |

training, we set AdamW (lr=1e-4), 150 epochs, and random masking with $k = 40\%$. The detailed preprocessing and experimental settings are shown in Appendix A.2 and A.3.

**MIMIC-IV-ECG**(Gow et al., 2023). The standard MIMIC-IV-ECG dataset contains 800,035 12-lead ECG from 161,352 distinct patients, sampled at 500 Hz for 10 seconds.

**PTB-XL**(Wagner et al., 2020). The standard PTB-XL dataset contains 21,799 12-lead ECG from 18,869 distinct patients, sampled at 500 Hz for 10 seconds. It can be used for two classification tasks. **Multi-label**: six sub-datasets (label counts): all (71), diagnostic (44), sub-diagnostic (24), super-diagnostic (5), rhythm (19), form (12). **Multi-class**: The "all" sub-dataset is employed, with samples reclassified into: normal, target diagnostic or other. For all tasks, each sub-dataset is partitioned into 8:1:1 using stratified split index originally assigned to the dataset (Nonaka & Seita, 2021). The model is trained and evaluated independently for five runs, and the final reported performance is the mean of the metrics calculated across all five test sets.

**G12EC**(Alday et al., 2020). The standard G12EC dataset contains 10,344 12-lead ECG from 10,344 distinct patients, sampled at 500 Hz for 10 seconds. The label count for multi-label is 30. We use multi-label stratification to evenly split the 30 class labels into 8:1:1. Then the model is trained and evaluated independently for five runs, and the final reported performance is the mean of the metrics calculated across all five test sets.

**VitalDB**(Lee et al., 2022). The data was obtained from non-cardiac (general, thoracic, urologic, and gynaecologic) surgery patients who underwent routine or emergency surgery. From this cohort, we select 2-lead ECG recordings (leads II and V5) from 3,390 cases, with an average duration of about 2 hours and a sampling rate of 500 Hz. For evaluation, the dataset is partitioned using five-fold cross-validation.

### 4.2 EVALUATION ON DOWNSTREAM CLASSIFICATION TASKS

The fundamental goal of learning ECG representations is to enable more accurate diagnostic performance. We evaluate classification under two settings: multi-label, predicting all diagnostic labels to assess overall diagnostic coverage, and multi-class, identifying the primary abnormality to assess categorization of core conditions. All experimental results are obtained under the pre-training setting on MIMIC-IV-ECG.

**Multi-label**. As summarized in Table 1, AtomECG establishes a new state-of-the-art in linear probing evaluation on six distinct sub-tasks across two benchmark datasets. Unlike competing approaches that depend on complex pre-processing, AtomECG achieves this superior performance without any data augmentation or waveform segmentation. The only exception to AtomECG's dominance is the rhythm task. We attribute to that HeartLang (Jin et al., 2025) employs an elaborate waveform segmentation, which is inherently optimized for rhythmic pattern analysis.

**Multi-class**. We select four common cardiac conditions for evaluation, which cover two core categories in ECG diagnosis: conduction abnormalities and arrhythmias. Table 2 presents the diagnostic performance of all eSSL models on both datasets under nonlinear probing. Here, we use a MLP(Linear-ReLU-Linear) as the classifier. AtomECG substantially outperforms all other eSSL models by a large margin.

### 4.3 INTERPRETABILITY ANALYSIS: FROM ATOMS TO PATHOPHYSIOLOGY

To investigate the contribution of atoms to high diagnostic performance, we analyze the statistically significant usage preferences for specific atoms and atom sequences across different patient cohorts from the PTB-XL, as summarized in Table 3. Our analysis cover 6 major and 12 minor diagnostic classes. We conduct a chi-squared ($\chi^2$) test of independence for plenty of atoms and atom subsequences and report only those with p-value<0.05, ensuring that the identified patterns are distinctively preferred and not due to random chance.

Our findings reveal that the learned codebook effectively captures class-specific morphological patterns. Each major diagnostic category is characterized by a unique set of atoms. For instance, atoms 347, 494, 813 are consistently present across all sub-classes of *CD*, while atoms 52, 357, 535 are identified as significant markers for *HYP*. At the same time, each sub-class contains several unique subsequences. These statistically validate, class-exclusive atoms and sequences serve as potent, discrete features for classification, providing a transparent basis for diagnoses.

Furthermore, the learned associations between different disease cohorts provide a powerful, data-driven validation of the model's understanding of cardiac pathophysiology. For example, atom 357 is found to be a significant shared marker for both *CD-LBBB* and *HYP*. This discovery aligns perfectly with: chronic pressure or volume overload leading to cardiac hypertrophy often causes structural changes, such as stretching and fibrosis, that can damage the heart's electrical conduction pathways, frequently manifesting as *LBBB*. The model's identification of a shared atom for these conditions indicates it has captured this underlying pathological link, not just superficial waveform similarities. Similarly, atom 113 is exhibits a significant co-occurrence in both *MI* and *STTC*. This reflects their shared ischemic etiology. ST-T abnormalities are a hallmark electrical manifestation of myocardial ischemia, the very process that results in infarction when severe and prolonged. By assigning a common atom, AtomECG demonstrates its capacity to learn fundamental pathological signatures, moving beyond simple pattern matching to grasp clinically meaningful abstractions.

### 4.4 VALIDATING ECG GRAMMAR VIA NEXT-ATOM PREDICTION

To verify that the learned atoms follow a predictable compositional structure, we design a next-atom prediction task. We train an GPT on atom sequences generated by AtomECG, which is pre-trained on MIMIC-IV-ECG and fine-tuned on PTB-XL. We then compare two predictive approaches: (1) predicting the next discrete index and then decoding it into the signal, and (2) directly predicting the next signal segment. As shown in Table 4, our "AtomECG+GPT" method outperforms all competing methods. This result confirms that the learned atoms form a high-quality, structured representation whose sequences adhere to a predictable grammar, making sequence-level modeling in the discrete space more effective than in the continuous domain. Figure 2 further illustrates the model's strong predictive performance on signal sequences of length 250 and 500.

Table 3: Statistically significant atoms and atom subsequences associated with different diagnostic classes in PTB-XL ($\chi^2$ test, p<0.05). [1]Conduction Disturbance, [2]Hypertrophy, [3]Myocardial Infarction, [4]ST/T Change, [5]Proportion of individuals containing the common subsequence.

| | Subclass | Description | Total | Atom usage(ratio[5]) |
|---|---|---|---|---|
| **NORM** | NORM | Normal ECG | 9528 | 1,24,126,170,269,351,395,542, 696,722,785,822,867,965,1010. 24-696-701-110(86%) . . . |
| **CD**[1] | RBBB | Right bundle branch block | 1660 | 77,79,346,347,494,497,813. 346-357-77-494(73%) . . . |
| | LBBB | Left bundle branch block | 613 | 80,150,346,347,357,494,497,813. 346-357-80-813-494(72%) . . . |
| | AVB | Atrioventricular block | 827 | 2,111,333,347,494,813,815. 111-2-813-494(81%) . . . |
| **HYP**[2] | LVH | Left ventricular hypertrophy | 2137 | 52,80,150,357,535,778. 778-357-535(74%) . . . |
| | SEHYP | Septal hypertrophy | 30 | 22,52,346,357,535,813. 346-357-535(80%) . . . |
| **MI**[3] | AMI | Anterior myocardial infarction | 354 | 113,334,425,436,479,1021. 436-113-1021(76%) . . . |
| | IMI | Inferior myocardial infarction | 2685 | 52,81,113,279,425,743,905,1001. 113-905-425(84%) . . . |
| **STTC**[4] | ISCA | Ischemic in anterior leads | 1016 | 2,112,113,456,921. 113-2-456-921(70%) . . . |
| | STTC | ST-T changes | 2329 | 52,112,113,333,660,743,921,980,999. 113-743-52(79%) . . . |
| **Rhythm** | AFIB | Atrial fibrillation | 1514 | 67,74,162,293,343,586,712,798,980. 798-712-67-586(95%) . . . |
| | STACH | Sinus tachycardia | 826 | 6,28,67,333,586,712,798,980. 798-712-67-586(96%) . . . |

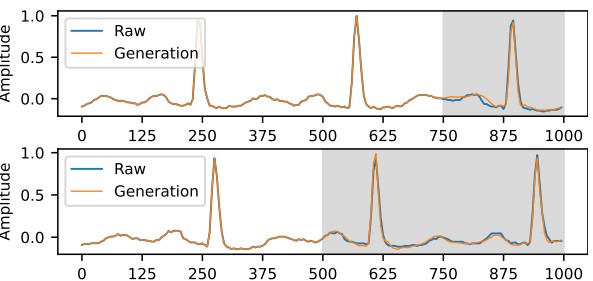

Figure 2: Next-atom prediction with 250-signal-length and 500-signal-length on PTB-XL lead II.

Table 4: Performance of next-atom prediction on PTB-XL(500-signal-length).

| Model | MAE | RMSE |
|---|---|---|
| *Signal prediction* | | |
| TimeGPT | 0.274 | 0.299 |
| Autoformer | 0.256 | 0.257 |
| *Index prediction* | | |
| Random+GPT | 0.856 | 0.894 |
| HeartLang+GPT | 0.280 | 0.292 |
| **AtomECG+GPT** | **0.054** | **0.061** |

## 4.5 SCALABILITY AND GENERALIZATION ANALYSIS

**Long-Term Dynamic Monitoring**. While preceding experiments focused on short-term (10s) ECG analysis, we hypothesize that the atomic paradigm of AtomECG is particularly advantageous for long-term recordings. To validate this, we evaluate the long-term atom prediction on VitalDB, which is designed to assess two key aspects: (1) the accuracy of atom representations over extended time horizons, and (2) the robustness of the learned ECG grammar in long-term sequences. As shown in Table 5 for the next-atom prediction on 2000-signal-length, AtomECG demonstrates superior predictive performance, outperforming all competing models and substantiating its suitability for continuous monitoring. The visualization of its long-term prediction is provided in Figure 3.

The model's proficiency in long-term analysis provides a strong foundation for monitoring applications, particularly for the emergency warning of sudden cardiac events. We illustrate this with a case study of an epinephrine injection event from VitalDB. Figure 4 contrasts the atom usage frequency in the 10-second intervals immediately before and after the injection. We observe a marked increase

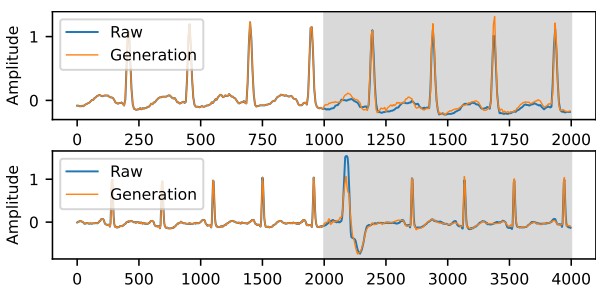

Figure 3: Next-atom prediction with 1000-signal-length and 2000-signal-length on VitalDB lead II.

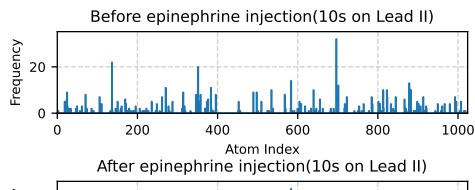

Figure 4: Atom usage frequency before and after epinephrine injection.

Table 5: Performance of next-atom prediction on VitalDB(2000-signal-length).

| Model | MAE | RMSE |
|---|---|---|
| *Signal prediction* | | |
| TimeGPT | 0.605 | 0.625 |
| Autoformer | 0.594 | 0.642 |
| *Index prediction* | | |
| Random+GPT | 0.905 | 0.926 |
| HeartLang+GPT | 0.617 | 0.655 |
| **AtomECG+GPT** | **0.122** | **0.143** |

Table 6: Cross-dataset generalization of codebook atoms: next-atom prediction performance on PTB-XL lead II (500-signal-length) under different pre-training datasets and fine-tuning settings.

| Few-shot | Pre-train on MIMIC | | Pre-train on VitalDB | |
|---|---|---|---|---|
| | MAE | RMSE | MAE | RMSE |
| ✗ | 0.086 | 0.089 | 0.099 | 0.108 |
| 1% | 0.075 | 0.076 | 0.067 | 0.072 |
| 10% | 0.056 | 0.066 | 0.058 | 0.064 |
| full | 0.049 | 0.057 | 0.053 | 0.058 |

in the usage of atoms strongly associated with sinus tachycardia in the post-injection window. This finding is perfectly aligned with the known clinical effect of epinephrine as a potent cardiac stimulant, showcasing AtomECG's ability to detect acute significant changes in cardiac state.

**Cross-Dataset Generalization**. If ECG atoms are fundamental building blocks, their grammar should be universal and transferable across different patient populations. To test this, we conduct a zero-shot evaluation, pre-training AtomECG on either VitalDB or MIMIC-IV-ECG and then applying it directly to PTB-XL without any fine-tuning. As reported in Table 6, the model demonstrates remarkable predictive accuracy in all few-shot settings. This strong performance confirms that AtomECG learns a robust and universal grammar that captures fundamental physiological patterns rather than dataset-specific artifacts, underscoring its potential for reliable clinical application.

**Ablation Studies**. The two most critical hyperparameters in AtomECG are the atom granularity (determined by patch size and stride) and the codebook size (number of atoms). Our guiding principle is to find a balance: each atom, as a building block, should be simple enough to be fundamental and reusable, yet not so granular that it models pure noise. Based on this principle, we choose the patch size of 16 with a stride of 8 and the codebook of 1024, ensuring a rich yet efficiently used set of building blocks. Also, our ablations show that the grammar manifold improves codebook structure, while geometric alignment enhances atom expressiveness, confirming the effectiveness of Two-Scale Manifold Alignment. The detailed ablation studies are shown in Appendix A.5.

## 5 CONCLUSION

In this work, we introduce AtomECG, a novel eSSL framework that models ECGs as sequences of discrete atoms governed by an underlying grammar. By moving beyond coarse, human-defined features, AtomECG learns the fundamental building blocks of cardiac signals. Leveraging Two-Scale Manifold Alignment, it achieves SOTA across diverse diagnostic tasks while offering strong interpretability. Also, AtomECG shows robustness in long-term monitoring and cross-population generalization, showing promise for continuous health tracking and personalized medicine. By combining high performance with clinical interpretability, AtomECG contributes to more reliable and transparent ECG analysis.

ETHICS STATEMENT

This work uses only publicly available, de-identified ECG datasets released with appropriate ethical approvals and data use agreements. No additional human data collection was performed. While our methods aim to advance ECG representation learning for research and clinical applications, we caution against inappropriate deployment without medical oversight and note that dataset biases may influence results. The authors declare no conflicts of interest.

REPRODUCIBILITY STATEMENT

We have made extensive efforts to ensure the reproducibility of our work. The architecture details, training hyperparameters, and evaluation protocols are described in Section 4.1 and Appendix A.3. The datasets used in our experiments are publicly available, and the preprocessing steps are detailed in Appendix A.2. Our source code is included in the supplementary files.

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

# A APPENDIX

## A.1 METHOD DETAILS

### A.1.1 DEOCDER

As the primary component of AtomECG decoder, the computation of S4D(Gu et al., 2022) is shown in Algorithm 2.

---

**Algorithm 2** S4D Computation in AtomECG Decoder

---

**Require**: Feature $\mathbf{g} \in \mathbb{R}^{c \times l}$.
**Require**: SSM Parameters $(\mathbf{A}, \mathbf{C}, \mathbf{\Delta t}, \mathbf{D})$.

1: $\tilde{\mathbf{C}} = \mathbf{C} \odot (\exp(\mathbf{A} \odot \mathbf{\Delta t}) - \mathbf{1}) \oslash \mathbf{A}$
2: **for** $i \leftarrow 0$ **to** $l - 1$
3:     $\mathbf{k}_{:,i} = 2 \cdot \mathrm{Re}\left(\sum_n \tilde{\mathbf{C}}_{:,n} \odot \exp(\mathbf{A}_{:,n} \odot \mathbf{\Delta t} \cdot i)\right)$
4: **end for**
5: $\mathbf{k}_{\mathrm{fft}} \leftarrow \mathrm{RFFT}(\mathbf{k}, n = 2l)$, $\mathbf{g}_{\mathrm{fft}} \leftarrow \mathrm{RFFT}(\mathbf{g}, n = 2l)$
6: $\mathbf{y} \leftarrow \mathrm{iRFFT}(\mathbf{g}_{\mathrm{fft}} \odot \mathbf{k}_{\mathrm{fft}})[:, :l]$
7: $\hat{\mathbf{x}}^{c \times l} = \underset{c \to c}{\mathrm{Linear}}(\mathrm{GELU}(\mathbf{y} + \mathbf{g} \odot \mathbf{D}))$
8: **Return** $\hat{\mathbf{x}}$

---

- (Lines 1-4): A convolutional kernel $\mathbf{k}$ of length $l$ is generated from the SSM parameters $(\mathbf{A}, \mathbf{C}, \mathbf{\Delta t}, \mathbf{D})$, embedding the model's dynamics into the kernel.

- (Lines 5-6): Convolution is performed via Fast Fourier Transform(FFT), where the input $\mathbf{g}$ and kernel $\mathbf{k}$ are transformed into the frequency domain, multiplied element-wise, and then mapped back to the time domain with an inverse FFT. RFFT denotes the FFT for real-valued inputs.

- (Line 7): The result of the convolution is added to a skip connection controlled by the parameter $\mathbf{D}$. Finally, the output undergoes a GELU activation followed by a linear layer.

### A.1.2 DETAILED ANALYSIS OF TWO-SCALE MANIFOLD ALIGNMENT

This appendix provides a detailed theoretical justification for the Two-Scale Manifold Alignment (TSMA) mechanism introduced in the main paper. Our goal is to dissect the limitations of standard vector quantization for periodic signals like the ECG and to formally describe how TSMA's global and local components synergize to overcome these challenges.

**The Problem: Contextual Ambiguity in Quantization of Periodic Signals**

VQVAE (van den Oord et al., 2017) aims to map a continuous vector $\mathbf{z} \in \mathbb{R}^d$ to a discrete vector $\mathbf{c}_k$ from a finite codebook $\mathcal{C} = \{\mathbf{c}_1, \ldots, \mathbf{c}_N\}$. The standard approach finds the nearest neighbor in Euclidean space:

$$Q(\mathbf{z}) = \mathbf{c}_{k^*}, \quad \text{where } k^* = \underset{k}{\arg\min} \|\mathbf{z} - \mathbf{c}_k\|_2^2. \tag{5}$$

This operation partitions the continuous space $\mathbb{R}^d$ into a Voronoi tessellation, where each cell corresponds to the basin of attraction for a codebook vector $\mathbf{c}_k$.

While effective for many applications, this formulation is fundamentally context-agnostic. The assignment of $\mathbf{z}$ depends solely on its position, irrespective of the temporal context from which it was derived. For a periodic and highly structured signal like an ECG, this leads to a critical information bottleneck. Consider the encoder output for a patch, $\mathbf{r}_j$. The nearest-neighbor rule cannot distinguish between:

(1) A patch from a healthy QRS complex in a regular sinus rhythm.

(2) A patch from a morphologically similar QRS complex that is part of a persistent pathology (e.g., bundle branch block), where the surrounding context (e.g., T-wave morphology) is altered.

(3) A patch from a morphologically similar QRS complex occurring at an anomalous time (e.g., a Premature Ventricular Contraction, PVC), where the global temporal context is violated.

In all three cases, the local morphology of the patch might be nearly identical, causing them to be mapped to the same atom $\mathbf{c}_k$. This collapses distinct physiological states into a single discrete symbol, at the cost of essential diagnostic information.

The problem is exacerbated during backpropagation. The Straight-Through Estimator (STE), a common technique for training VQ models, approximates the gradient of the non-differentiable quantization step with an identity mapping: $\nabla_{\mathbf{z}} L \approx \nabla_{Q(\mathbf{z})} L$. This implies that for any two encoder outputs $\mathbf{r}_i$ and $\mathbf{r}_j$ that are mapped to the same code vector $\mathbf{c}_k$, the gradient passed back to the encoder is identical:

$$\text{if } Q(\mathbf{r}_i) = Q(\mathbf{r}_j) = \mathbf{c}_k, \quad \text{then } \nabla_{\mathbf{r}_i} L \approx \nabla_{\mathbf{c}_k} L \text{ and } \nabla_{\mathbf{r}_j} L \approx \nabla_{\mathbf{c}_k} L. \tag{6}$$

This uniform gradient signal provides no incentive for the encoder to learn representations that separate subtly different but contextually distinct patches within the same Voronoi cell. TSMA is designed to resolve this very issue.

**Global Scale: Structuring the Codebook via a Learnable Manifold**

To imbue the quantization process with global context, we first address the structure of the codebook itself. A static codebook, typically updated via exponential moving averages or a commitment loss, learns to represent high-density regions of the feature space but lacks an explicit mechanism to enforce relationships between atoms that reflect the underlying grammar of the signal.

Refer to (Zhu et al., 2024), we introduce a learnable linear transformation $\mathbf{E} \in \mathbb{R}^{d \times d}$ to reparameterize the codebook space. Instead of matching an encoder output $\mathbf{r}_j$ to a static code $\mathbf{c}_n \in \mathcal{C}$, we match it to a dynamically transformed code $\mathbf{c}_n \mathbf{E}$. The quantization rule becomes:

$$n_j^* = \arg\min_n \|\mathbf{r}_j - \mathbf{c}_n \mathbf{E}\|_2^2. \tag{7}$$

This transformation maps the discrete set of codebook vectors $\mathcal{C}$ onto a learnable linear submanifold, which we term the "grammar manifold," $\mathcal{M}_{\mathcal{C}} = \{\mathbf{c}\mathbf{E} \mid \mathbf{c} \in \mathcal{C}\}$. The matrix $\mathbf{E}$ acts as a set of shared parameters that defines the geometry of this manifold.

The key advantage arises during optimization. The gradient of the loss $L$ with respect to $\mathbf{E}$ aggregates information from all selected atoms within a mini-batch:

$$\nabla_{\mathbf{E}} L = \sum_{j \in \text{batch}} \nabla_{\mathbf{E}} L(\mathbf{r}_j, \mathbf{c}_{n_j^*} \mathbf{E}). \tag{8}$$

This collective update mechanism has two profound effects:

(1) **Shared Structure Learning**: Since every selected patch contributes to the update of $\mathbf{E}$, the transformation learns to arrange the atoms on the manifold $\mathcal{M}_{\mathcal{C}}$ in a way that optimally represents the global distribution of ECG morphologies. It learns the "principal axes" of the ECG grammar, ensuring that the relationships between atoms (e.g., proximity on the manifold) are semantically meaningful.

(2) **Prevention of Codebook Collapse**: In standard VQ, unselected code vectors receive no update signal. Here, even though the base codes $\mathbf{c}_n$ are fixed, their effective positions $\mathbf{c}_n \mathbf{E}$ are all jointly optimized through $\mathbf{E}$. A gradient signal from matching patch $\mathbf{r}_j$ to atom $\mathbf{c}_{n_j^*}$ will update $\mathbf{E}$, thereby shifting the positions of all other atoms $\mathbf{c}_n \mathbf{E}$ (for $n \neq n_j^*$). This ensures that the entire codebook co-evolves, promoting higher atom utilization and a more structured latent space.

In essence, the global component learns an optimal linear subspace for the ECG atoms to reside in, ensuring the vocabulary as a whole is coherent and grammatically structured.

**Local Scale: Geometry-Aware Gradient Steering with Householder Transformations**

While the global manifold provides a structured target space, we still require a mechanism to map individual patches to it in a context-sensitive manner. As established, STE is insufficient because its

gradient is insensitive to the geometric relationship between an encoder output $\mathbf{r}_j$ and its target code $\mathbf{q}_j\mathbf{E}$, where $\mathbf{q}_j = \mathbf{c}_{n_j^*}$.

Refer to (Fifty et al., 2024), we introduce a more informative gradient pathway using a Householder transformation. The objective is to define a differentiable operation that transforms $\mathbf{r}_j$ into $\mathbf{q}_j\mathbf{E}$, such that the gradient of this operation encodes the geometric discrepancy (both in magnitude and direction) between the two vectors.

Let $\mathbf{u} = \frac{\mathbf{q}_j\mathbf{E}}{\|\mathbf{q}_j\mathbf{E}\|_2}$ and $\mathbf{v} = \frac{\mathbf{r}_j}{\|\mathbf{r}_j\|_2}$ be the unit vectors corresponding to the target and the encoder output, respectively. A Householder transformation is a reflection about a hyperplane. To map $\mathbf{v}$ to $\mathbf{u}$, we must reflect $\mathbf{v}$ across the hyperplane orthogonal to the vector $\mathbf{w} = \mathbf{u} - \mathbf{v}$. The Householder matrix representing this reflection is:

$$\mathbf{H} = \mathbf{I} - 2\frac{\mathbf{w}\mathbf{w}^T}{\mathbf{w}^T\mathbf{w}}. \tag{9}$$

This matrix is orthogonal and symmetric ($\mathbf{H} = \mathbf{H}^T = \mathbf{H}^{-1}$) and satisfies $\mathbf{H}\mathbf{v} = \mathbf{u}$. To account for the difference in magnitude, we introduce a scaling factor. The complete transformation matrix $\mathbf{R}_{\theta,j}$ is thus:

$$\mathbf{R}_{\theta,j} = \frac{\|\mathbf{q}_j\mathbf{E}\|_2}{\|\mathbf{r}_j\|_2}\mathbf{H} = \frac{\|\mathbf{q}_j\mathbf{E}\|_2}{\|\mathbf{r}_j\|_2}\left(\mathbf{I} - 2\frac{(\mathbf{u}-\mathbf{v})(\mathbf{u}-\mathbf{v})^T}{(\mathbf{u}-\mathbf{v})^T(\mathbf{u}-\mathbf{v})+\epsilon}\right). \tag{10}$$

By construction, $\mathbf{R}_{\theta,j}\mathbf{r}_j = \mathbf{q}_j\mathbf{E}$. In the forward pass, the quantized representation is $\hat{\mathbf{f}}_j = \mathbf{q}_j\mathbf{E}$. To enable backpropagation, we define it as $\hat{\mathbf{f}}_j = \mathrm{sg}[\mathbf{R}_{\theta,j}]\mathbf{r}_j + \mathrm{sg}[\mathbf{q}_j\mathbf{E} - \mathbf{R}_{\theta,j}\mathbf{r}_j]$, which is equivalent to STE in the forward pass but uses a different gradient estimator. The gradient flowing back to the encoder output $\mathbf{r}_j$ is:

$$\nabla_{\mathbf{r}_j}L = \mathbf{R}_{\theta,j}^T\nabla_{\hat{\mathbf{f}}_j}L. \tag{11}$$

This is the central mechanism for local alignment. Unlike the identity matrix of STE, $\mathbf{R}_{\theta,j}$ is an input-dependent linear operator. The gradient $\nabla_{\hat{\mathbf{f}}_j}L$ is rotated and scaled by $\mathbf{R}_{\theta,j}^T$. The magnitude of the rotational component is proportional to the angle between $\mathbf{r}_j$ and $\mathbf{q}_j\mathbf{E}$.

- If $\mathbf{r}_j$ is perfectly aligned with $\mathbf{q}_j\mathbf{E}$ (i.e., $\mathbf{u} = \mathbf{v}$), then $\mathbf{w} = 0$ and $\mathbf{R}_{\theta,j}$ becomes a simple scaling matrix. The gradient is scaled but not rotated.
- If $\mathbf{r}_j$ and $\mathbf{q}_j\mathbf{E}$ are poorly aligned, the off-diagonal elements of $\mathbf{R}_{\theta,j}$ are large, inducing a significant rotational component in the gradient. This corrective torque pushes the encoder to produce representations that are not just closer in distance but also better aligned in direction.

This geometry-aware gradient is precisely what is needed to resolve local ambiguities. A patch from a PVC, though close in Euclidean distance to a normal QRS atom, has a significant angular deviation due to its different context being encoded. The Householder-based gradient will penalize this angular mismatch, forcing the encoder to map it to a more appropriate, distinct atom.

**Local Scale: The Role of Angular Information in Representing ECG Context**

A central claim of our local alignment mechanism is that the angle between an encoder output $\mathbf{r}_j$ and its target atom $\mathbf{q}_j\mathbf{E}$ is a meaningful signal. Here, we provide a theoretical justification for why angular deviation in the latent space is a natural and effective proxy for contextual variation in ECG signals.

An encoder, particularly a deep one with a large receptive field (such as the Transformer used in AtomECG), does not operate on a patch in isolation. The representation for the $j$-th patch, $\mathbf{r}_j$, is a function of the entire input signal $\mathbf{x}$, i.e., $\mathbf{r}_j = \mathrm{Encoder}(\mathbf{x})_j$. Consequently, $\mathbf{r}_j$ is tasked with embedding not only the local morphology of the patch itself but also its surrounding temporal and physiological context.

We posit that the encoder learns to structure the latent space such that vectors corresponding to a specific morphological class (e.g., all "QRS-like" patches) lie in the vicinity of a "prototype" direction on the grammar manifold. The primary direction of a vector $\mathbf{r}_j$ thus encodes its dominant

morphological characteristic. For instance, let $\mathcal{V}_{\text{QRS}}$ be the subspace or cone in $\mathbb{R}^d$ containing representations of all QRS-like morphologies.

Contextual information (such as whether a QRS complex is part of a healthy sinus rhythm, a persistent arrhythmia, or a sporadic ectopic beat) acts as a modulator on this base representation. A powerful and efficient way for the network to encode this modulation is to apply a rotation to the base vector. A small rotation signifies a subtle contextual shift (e.g., slight rate variability), while a large rotation signifies a major contextual anomaly (e.g., a PVC appearing where a P-wave is expected).

We can formalize this intuition. Consider a representation $\mathbf{r}_j$ and its target on the manifold $\mathbf{m}_j = \mathbf{q}_j \mathbf{E}$. We can decompose $\mathbf{r}_j$ into a component parallel to $\mathbf{m}_j$ and a component orthogonal to it:

$$\mathbf{r}_j = \mathbf{r}_{j,\parallel} + \mathbf{r}_{j,\perp}, \tag{12}$$

where $\mathbf{r}_{j,\parallel} = \text{proj}_{\mathbf{m}_j}(\mathbf{r}_j)$ and $\mathbf{r}_{j,\perp}$ is the orthogonal residual. The magnitude $\|\mathbf{r}_{j,\parallel}\|$ can be interpreted as encoding the strength or confidence of the primary morphology, while the orthogonal component $\mathbf{r}_{j,\perp}$ can be seen as encoding the contextual deviation. The angle $\theta$ between $\mathbf{r}_j$ and $\mathbf{m}_j$ is given by $\cos(\theta) = \frac{\mathbf{r}_j \cdot \mathbf{m}_j}{\|\mathbf{r}_j\| \|\mathbf{m}_j\|}$, which is directly related to the relative magnitude of the orthogonal component: $\|\mathbf{r}_{j,\perp}\| = \|\mathbf{r}_j\| \sin(\theta)$.

Therefore, a larger angular deviation implies a more significant contextual discrepancy relative to the prototype morphology defined by the atom on the manifold. The standard STE gradient, being an identity mapping, is blind to this orthogonal component and provides a learning signal that only minimizes the Euclidean distance, potentially collapsing $\mathbf{r}_{j,\perp}$ to zero and destroying the encoded contextual information.

In contrast, the Householder-based gradient, $\nabla_{\mathbf{r}_j} L = \mathbf{R}_{\theta,j}^T \nabla_{\hat{\mathbf{f}}_j} L$, is explicitly designed to act on this angular discrepancy. The rotational nature of $\mathbf{R}_{\theta,j}$ ensures that the gradient provides a corrective torque that is proportional to the angular error. This forces the encoder to learn meaningful rotations, preserving and refining the representation of context, which is critical for distinguishing clinically significant events that are defined not just by their shape but by their timing and surroundings.

**Synergy: Fusing Global Structure and Local Precision**

The global and local scales of TSMA are not independent but deeply synergistic. The global grammar manifold $\mathcal{M}_{\mathcal{C}}$ provides a stable, well-structured, and semantically meaningful target space for the atoms. It prevents the codebook from becoming a disorganized collection of cluster centroids and instead shapes it into a coherent representation of the ECG language. The local geometric alignment mechanism acts as a high-precision mapping function onto this manifold. It provides the encoder with rich, geometry-aware feedback, enabling it to learn subtle yet diagnostically crucial morphological distinctions.

**Residual Quantization**

While a single quantization step using TSMA provides a robust mapping, its expressive power is inherently limited by the size of the codebook $N$. A single atom must approximate the encoder output $\mathbf{r}_j$, and the quantization error, $\boldsymbol{\epsilon}_j = \mathbf{r}_j - \mathbf{q}_j \mathbf{E}$, represents the information lost in this approximation. For complex signals like ECG, this residual error can contain substantial fine-grained morphological details that are diagnostically relevant.

To capture this information, we employ a multi-stage, hierarchical approach known as Residual Quantization (Zeghidour et al., 2021). We define the process formally for $h$ levels of quantization, each with its own codebook $\mathcal{C}^{(i)}$ and grammar manifold transformation $\mathbf{E}^{(i)}$. The initial input to the process is the encoder output, which we denote as the first-level residual: $\mathbf{r}^{(0)} = \mathbf{f}$. For each level $i$ from 0 to $h-1$, we perform the following steps for every patch $j$:

(1) **Quantize the current residual**: We apply the full TSMA procedure to the residual $\mathbf{r}_j^{(i)}$ using the $i$-th codebook and transformation matrix. This involves finding the best matching atom $\mathbf{q}_j^{(i)} = \mathbf{c}_{n_j^*}^{(i)}$ on the $i$-th manifold and generating the quantized representation $\hat{\mathbf{f}}_j^{(i)}$ for

backpropagation using the corresponding Householder matrix $\mathbf{R}_{\theta,j}^{(i)}$:

$$n_j^* = \arg\min_n \|\mathbf{r}_j^{(i)} - \mathbf{c}_n^{(i)}\mathbf{E}^{(i)}\|_2^2, \tag{13}$$

$$\hat{\mathbf{f}}_j^{(i)} = \text{sg}[\mathbf{R}_{\theta,j}^{(i)}]\mathbf{r}_j^{(i)}. \tag{14}$$

(2) **Compute the next residual**: The new residual for the next stage is the error from the current stage's quantization, calculated in the transformed space:

$$\mathbf{r}_j^{(i+1)} = \mathbf{r}_j^{(i)} - \mathbf{q}_j^{(i)}\mathbf{E}^{(i)}. \tag{15}$$

After $h$ stages, the final quantized representation of the original patch $\mathbf{f}_j$ is the sum of the quantized vectors from each level:

$$\hat{\mathbf{f}}_{\text{total},j} = \sum_{i=0}^{h-1} \hat{\mathbf{f}}_j^{(i)}. \tag{16}$$

This summation is an approximation of the original encoder output: $\mathbf{f}_j \approx \sum_{i=0}^{h-1} \mathbf{q}_j^{(i)}\mathbf{E}^{(i)}$.

Residual quantization enhances fidelity by recursively encoding residual errors, yielding a coarse-to-fine representation. Unlike single-stage quantization, which discards fine-grained details, this refinement enables closer approximation of encoder outputs and is critical for capturing the complex morphologies of ECG signals.

## A.2 DATASETS PREPROCESSING AND METRIC

**Pre-train Preprocessing**. The preprocessing pipeline sequentially applies NaN/Inf value handling, amplitude outlier exclusion (threshold$\pm 11$), statistical filtering (mean$\in[-1.5, 1.5]$, variance$\in[0.05, 2]$), and temporal variation filtering (threshold 10 units for adjacent differences).

**Metric**.

- Macro AUROC: In multi-label classification, the area under the receiver operating characteristic curve is computed for each class independently by varying the classification threshold, and the final score is obtained by averaging the per-class AUCs.

- Macro F1: Computes the harmonic mean of precision and recall for each class individually, then averages these F1 scores.

- Perplexity: Quantifies the uniformity of vector usage within a codebook. Defined as: Perplexity $= \exp\left(-\sum_{c=1}^{K} p(c)\log p(c)\right)$. Here, $\sum_{c=1}^{K} p(c)\log p(c)$ represents the entropy of the codebook distribution. Higher perplexity indicates greater uncertainty in vector selection. When vectors are uniformly used ($p(c) = 1/K$), perplexity equals the codebook size $K$. Conversely, concentrated usage reduces the perplexity.

## A.3 IMPLEMENTATION DETAILS

**Pre-training settings**. All experiments were conducted on a NVIDIA H200 GPU equipped with 141GB memory. The performance of all baselines and eSSL models was evaluated under consistent preprocessing, strictly following their publicly available code and training configurations. During pre-training on MIMIC-IC-ECG, 95% is used for training and 5% for validation, where the validation set is used to monitor training and prevent codebook collapse. For training on VitalDB, we segment each patient's data into 1-minute clips, with a processing window of 1 minute. All other procedures remain identical to those used for MIMIC-IV-ECG. The overall architecture of Atom-ECG comprises three main components: an encoder, a quantization module, and a decoder. The encoder architecture draws inspiration from PatchTST (Nie et al., 2023). The detailed hyperparameters are as follows:

- Encoder: Number of layers: m = 6
- Encoder: Number of attention heads: 4

- Encoder: Hidden dimension (d_model): 128
- Encoder: Feed-forward network dimension: 512
- Quantization: Residual layer: h = 4
- Decoder: Number of S4D layers: 4
- Decoder: State-space dimension: 64
- Decoder: Dropout rate: 0.1
- Decoder: Minimum and maximum values for time step: 0.001 and 0.1
- Training on MIMIC-IV-ECG: Epoch-150, Optimizer-AdamW-1e-4, Batch-128, Masking rate-40%
- Training on VitalDB: Epoch-100, Optimizer-AdamW-5e-5, Batch-64, Masking rate-40%

The loss function in pre-training can be written as $\mathcal{L} = \mathrm{MSE}(\mathbf{x}, \hat{\mathbf{x}}) + \mathcal{L}_{\mathrm{vq}} + \mathcal{L}_{\mathrm{commit}}$, where

$$\mathcal{L}_{\mathrm{vq}} = \sum_{i=0}^{h-1} \left( \alpha_i \|\mathrm{sg}(\mathbf{r}^{(i)}) - \mathbf{Q}^{(i)}\mathbf{E}^{(i)}\|_2^2 \right), \quad \mathcal{L}_{\mathrm{commit}} = \sum_{i=0}^{h-1} \left( \beta_i \|\mathbf{r}^{(i)} - \mathrm{sg}(\mathbf{Q}^{(i)}\mathbf{E}^{(i)})\|_2^2 \right).$$

And for simplicity, we set the vq and commitment weights to $\alpha = \beta = 0.25$.

**Downstream classification**. Different classification tasks require different learning rates. We report the choices for multi-label and multi-class settings under linear and nonlinear probing.

Table 7: The learning rate on multi-label classification tasks under linear probing.

| all | diag. | sub. | super. | form | rhythm | G12EC |
|-----|-------|------|--------|------|--------|-------|
| 1e-4 | 1e-4 | 1e-4 | 5e-5 | 1e-5 | 5e-5 | 1e-4 |

Table 8: The learning rate on multi-class classification tasks under nonlinear probing.

| AF | I-AVB | LBBB | RBBB | AF | I-AVB | LBBB | RBBB |
|----|-------|------|------|-----|-------|------|------|
| | *lr on PTB-XL* | | | | *lr on G12EC* | | |
| 4e-5 | 5e-5 | 5e-5 | 5e-5 | 8e-5 | 6e-5 | 6e-5 | 6e-5 |

We adopt different probing strategies across the two tasks for the following reasons. For the multi-label task, linear probing is sufficient to compare the representational strength of different eSSL models. However, according to the supervised results from ECG baselines (Nonaka & Seita, 2021), no eSSL model consistently outperforms ResNet-18. Therefore, in the subsequent multi-class task, we additionally evaluate nonlinear probing. The results show that all eSSL models significantly surpass ResNet-18. We attribute this to the diverse diagnostic labels and their strong inter-label dependencies, where linear probing alone struggles to achieve accurate discrimination on small-scale datasets. Hence, we adopt different probing strategies for the two tasks, with the only difference being that the linear classifier is replaced by an MLP (Linear–ReLU–Linear).

**Downstream generation**. We compare against two main categories of models. The first category includes models that directly predict subsequent signal segments from a given preceding context. For this, we select strong baselines from the time-series forecasting domain, such as TimeGPT (Garza et al., 2023) and Autoformer (Wu et al., 2021). The second category comprises models with a predefined vocabulary, which perform prediction by forecasting the next token index. Among the baselines, only HeartLang (Jin et al., 2025) fits this category. For comparison, we also report the case where predictions are made by randomly selecting indices. For AtomECG, the workflow of the next-atom prediction is:

- Fine-tune on the downstream datasets;
- Freeze the encoder and codebook;
- Train a transformer decoder to model indices sequences. The hypeparameters are as follows: number of layers-6, number of attention heads-8, hidden dimension (d_model)-512, epoch-100, optimizer-AdamW-3e-4, Batch-64.

Given that we previously constructed a multi-layer codebook, here we only use the first-layer codebook for prediction. Post training, predicted indices retrieve corresponding atoms from the codebook, which are then decoded into the final ECG signal. And the decoder is from the fine-tuning stage.

## A.4 MORE EXPERIMENTS ON NEXT-ATOM PREDICTION

In the main text, we present the results of next-atom prediction with sequence lengths of 500 and 2000, and here we additionally report the results for lengths of 250 and 1000 in Figure 9 and Figure 10.

Table 9: Performance of next-atom prediction on PTB-XL(250-signal-length).

| Model | MAE | RMSE |
|---|---|---|
| *Signal prediction* | | |
| TimeGPT | 0.168 | 0.179 |
| Autoformer | 0.161 | 0.177 |
| *Index prediction* | | |
| Random+GPT | 0.814 | 0.875 |
| HeartLang+GPT | 0.138 | 0.149 |
| **AtomECG+GPT** | **0.022** | **0.025** |

Table 10: Performance of next-atom prediction on VitalDB(1000-signal-length).

| Model | MAE | RMSE |
|---|---|---|
| *Signal prediction* | | |
| TimeGPT | 0.443 | 0.475 |
| Autoformer | 0.415 | 0.434 |
| *Index prediction* | | |
| Random+GPT | 0.903 | 0.926 |
| HeartLang+GPT | 0.482 | 0.525 |
| **AtomECG+GPT** | **0.062** | **0.069** |

## A.5 ABLATION STUDIES

We conduct ablation studies along three dimensions. (i) Patch size. The patch length should be substantially shorter than a typical cardiac cycle to capture atomic morphologies, yet not too small to avoid noise. Table 11 reports results across different patch lengths and strides, where we ultimately adopt a length of 16 with a stride of 8. (ii) Atom size. We evaluate the effect of the number of atoms using perplexity as a metric for utilization. As shown in Table 12, 1024 atoms yield the most balanced usage, with each atom represented by a 128-dimensional embedding. (iii) Quantization scheme. Table 13 compares variants of the proposed Two-Scale Manifold Alignment. Using only manifold alignment produces a globally well-structured codebook but sacrifices sensitivity to subtle local pathologies. Using only rotation enables fine local matching but results in a disordered codebook lacking global grammar, leading to poor utilization and generalization. We further observe that residual learning of the codebook provides an additional accuracy gain.

Table 11: Ablation of patch size and stride during pre-training on MIMIC-IV-ECG.

| Patch Size | MSE | VQ-loss |
|---|---|---|
| *stride-8* | | |
| 8 | 0.00390 | 5.71e-5 |
| 16 | 0.00217 | 2.65e-5 |
| 24 | **0.00180** | 5.17e-4 |
| 32 | 0.00245 | 2.15e-3 |
| *stride-16* | | |
| 16 | 0.00762 | 2.01e-3 |
| 24 | 0.00567 | 1.24e-3 |
| 32 | 0.00564 | 1.03e-3 |
| *stride-32* | | |
| 32 | 0.01505 | 1.45e-3 |

Table 12: Ablation of codebook size during pre-training on MIMIC-IV-ECG.

| Atom Size | Perplexity | VQ-loss |
|---|---|---|
| *codebook-512* | | |
| 64 | 125/70/70/65 | 9.19e-5 |
| 128 | 113/101/85/79 | 8.75e-5 |
| 256 | 139/91/85/84 | 2.12e-4 |
| *codebook-1024* | | |
| 64 | 294/203/178/171 | 6.01e-5 |
| 128 | **317/200/185/187** | 2.65e-5 |
| 256 | 299/205/184/187 | 7.79e-5 |
| *codebook-2048* | | |
| 64 | 444/363/250/239 | 7.47e-5 |
| 128 | 606/300/288/202 | 7.05e-5 |
| 256 | 585/376/178/164 | 1.96e-4 |

Table 13: Ablation of the Two-scale Manifold Alignment on next-atom prediction(250-signal-length on PTB-XL).

| Quantization | Manifold | Rotation | Residual | MAE | RMSE | Perplexity |
|:---:|:---:|:---:|:---:|:---:|:---:|:---:|
| ✓ | - | - | - | 0.063 | 0.075 | 210 |
| ✓ | ✓ | ✓ | - | 0.032 | 0.034 | 425 |
| ✓ | - | ✓ | ✓ | 0.029 | 0.033 | $\left(\begin{smallmatrix} 225/176 \\ 145/135 \end{smallmatrix}\right)$ |
| ✓ | ✓ | - | ✓ | 0.045 | 0.047 | $\left(\begin{smallmatrix} 420/315 \\ 213/187 \end{smallmatrix}\right)$ |
| ✓ | ✓ | ✓ | ✓ | 0.022 | 0.025 | $\left(\begin{smallmatrix} 476/324 \\ 202/199 \end{smallmatrix}\right)$ |

## A.6 LIMITATION

Despite AtomECG's strong cross-population generalization demonstrated in our main experiments, we identify three key limitations that suggest avenues for future work.

First, the zero-shot generalization capability of AtomECG does not readily extend to highly specialized cohorts. We observe that the learned atoms and grammar struggle to adapt to out-of-distribution ECG patterns from athletes in a Norwegian dataset (Singstad, 2022), necessitating full fine-tuning to achieve comparable performance. This suggests that enhancing model generalization requires incorporating more diverse populations during pre-training. Furthermore, the model's robustness is challenged by signals with a low signal-to-noise ratio. Specifically, significant artifacts, such as those induced by intraoperative procedures in VitalDB, can disrupt the accurate identification of atoms and consequently compromise the reliability of the final analysis. Lastly, the current implementation of AtomECG has not been optimized for computational efficiency, hindering its direct deployment on resource-constrained edge devices. This is a critical barrier for real-world applications like early-warning systems in long-term Holter monitoring, which demand real-time, on-device processing. Future work should focus on optimizing the model's architecture to improve inference speed and enable practical deployment.

## A.7 THE USE OF LARGE LANGUAGE MODELS (LLMs)

We utilized a LLM to assist in the preparation of this manuscript. Its role was strictly limited to improving grammar, syntax, and clarity, with a primary focus on polishing the language in the Introduction section to ensure standard academic expression. The LLM was not used for generating core research ideas, developing the methodology, conducting the literature review, or interpreting results. All scientific contributions, analyses, and conclusions presented in this paper are entirely our own.

