# OpenReview forum: "Uncovering the Building Blocks of ECGs with a Discrete Autoencoder"
_ICLR.cc/2026/Conference — ICLR 2026 Conference Withdrawn Submission_

### Official Review · Reviewer_HN2Y · 2025-10-30

**Soundness:** 3
**Presentation:** 2
**Contribution:** 2
**Rating:** 2
**Confidence:** 4

**Summary:**

AtomECG introduces a novel electrocardiogram self-supervised learning (eSSL) framework that models ECG signals as sequences of discrete atoms governed by an underlying grammar. Unlike traditional approaches relying on coarse, human-defined features, AtomECG learns the fundamental building blocks of cardiac signals through a grammar-based representation. The framework leverages Two-Scale Manifold Alignment to achieve state-of-the-art performance across diverse diagnostic tasks while maintaining strong interpretability. Extensive experiments on PTB-XL and G12EC datasets demonstrate that AtomECG significantly outperforms existing contrastive and generative self-supervised learning methods in both multi-label and multi-class classification tasks. The framework shows robust performance in long-term monitoring and cross-population generalization, making it promising for continuous health tracking and personalized medicine applications.

**Strengths:**

1. AtomECG achieves state-of-the-art results across multiple diagnostic tasks on PTB-XL and G12EC datasets, significantly outperforming existing methods in both macro AUROC and macro F1 metrics.
2. The discrete atom representation provides inherent interpretability, which is a significant advantage for clinical adoption compared to black-box deep learning models.
3. Extensive evaluation across multiple tasks (multi-label and multi-class) and datasets (PTB-XL and G12EC) provides convincing evidence of the framework's effectiveness.

**Weaknesses:**

1. The atomic ECG representation learning has already been proposed in previous works like [1]. Though it is quite novel for the local/global view of VQ training, I cannot agree that the atomic perspective of ECG learning can be regarded as a contribution.
2. I believe the authors have misunderstood what an ablation study is. “The ablation study” presented for now is actually a kind of hyperparameter study. The contribution of individual components (particularly the Two-Scale Manifold Alignment) is not thoroughly analyzed through ablation studies, making it difficult to isolate the key innovations.
3. The notations within the paper are somewhat unclear. For example, what is c and l in line 148. Please make clear definition of each notation in the paper.
4. The paper does not analyze cases where AtomECG might fail or perform suboptimally, which is important for understanding the model's limitations in clinical settings. For example, the paper claims that the present methods “implicitly or explicitly, often operate at a coarse granularity dictated by human-defined fiducial points”, but in line 335, the authors, on the contrary, attribute the overperformance of the baseline to “an elaborate waveform segmentation”.
5. Please give a specific definition of ECG grammar and why the probing of GPT can be taken as the evaluation of ECG grammar learning. Otherwise, it cannot be taken as an effective evaluation. Also, please provide clear definitions of RMSE and MAE in this section. I wonder if it is the distance between the predicted and the GT embedding or the error of the codebook number.

[1] Hu, Y., Chen, J., Hu, L., Li, D., Yan, J., Ying, H., ... & Wu, J. (2024). Personalized heart disease detection via ecg digital twin generation. IJCAI.

**Questions:**

No more questions

---

### Official Review · Reviewer_jE8h · 2025-11-01

**Soundness:** 3
**Presentation:** 2
**Contribution:** 2
**Rating:** 2
**Confidence:** 5

**Summary:**

This paper introduces AtomECG, a new self-supervised learning framework for electrocardiogram (ECG) representation. AtomECG conceptualizes ECG signals as compositions of discrete morphological units (“ECG atoms”), each representing fundamental waveform patterns. It employs a Two-Scale Manifold Alignment mechanism to jointly align global grammar-level structure and local geometry-aware context. Experiments demonstrate that AtomECG outperforms prior ECG self-supervised methods across multiple diagnostic downstream tasks while offering interpretable, atom-level representations and strong generalization.

**Strengths:**

1.  **State-of-the-Art Performance:** AtomECG achieves state-of-the-art performance across a wide range of diagnostic tasks, outperforming numerous leading eSSL baselines. This empirical success, especially without requiring complex pre-processing or data augmentation, underscores the model's robust representation learning capabilities.
2.  **Demonstrated Clinical Interpretability:** The framework provides strong interpretability by explicitly linking learned atoms and atom sequences to pathological patterns. The identification of disease-specific atoms and shared markers between related conditions (e.g., atom 357 for CD-LBBB and HYP; atom 113 for MI and STTC) offers a transparent basis for diagnosis, potentially increasing clinical trust.

**Weaknesses:**

1.  **Ambiguous Motivation and Rationale for Quantization:**
*   The paper's narrative for introducing quantization lacks clarity. The transition from the limitations of existing eSSL (continuous, high-dimensional, opaque representations) to the need for ELP, and then to discrete quantization for overcoming human-defined waveform segmentation issues (attributed to ECGBERT, HeartLang) is not fully convincing. It's unclear why discrete representations are inherently superior for sub-waveform morphologies or how ECGBERT and HeartLang are strictly limited in this regard.
*   The unique advantages of quantization for ECG's periodic nature are not sufficiently emphasized in the introduction. As a result, Two-Scale Manifold Alignment (TSMA) appears to be the sole novelty, without a strong foundational argument for why a discrete autoencoder approach is fundamentally necessary for ECG beyond general VQ-VAE applications.
*   The ablation study for TSMA (Table 13) lacks a direct comparison against a naive or standard VQ-VAE quantization mechanism, making it difficult to fully appreciate TSMA's specific contributions to ECG representation learning.

2.  **Lack of Clarity in Methodology:**
*   The explanation of how Two-Scale Manifold Alignment effectively resolves the "ambiguity between local similarity and global cardiac context" across the three critical ECG scenarios (healthy periodic, consistent pathological, irregular pathological events) remains vague. While Algorithm 1 and Appendix A.1.2 provide mathematical details, a more intuitive, high-level explanation with illustrative examples is crucial for understanding its mechanism.

3.  **Inconsistencies and Insufficient Detail in Results:**
*   The different validation methods (linear probing for multi-label, nonlinear probing for multi-class) are mentioned in Appendix A.3, but the rationale and implications should be clearly discussed in the main Section 4.2 to ensure consistent understanding for the reader.
*   The absence of from scratch baseline performance for multi-label tasks (Table 1) makes it difficult to directly quantify the benefits of pre-training compared to supervised learning.
*   The claim that AtomECG achieves superior performance "without any data augmentation or waveform segmentation" unlike "competing approaches that depend on complex pre-processing" needs clarification. Many contrastive eSSL methods (e.g., CMSC) also operate without extensive data augmentation. The term "complex pre-processing" should be more precisely defined or justified.
*   The Next-atom prediction and Scalability and generalization analysis sections lack sufficient detail to fully understand the experiments' purpose and implications.
*   For next-atom prediction with overlapping patches, the specific method for evaluating performance (e.g., how the MAE/RMSE on signal reconstruction relate to discrete atom prediction accuracy) is not clearly outlined.
*   The epinephrine injection case study in long-term monitoring feels somewhat anecdotal and disconnected from the broader discussion of long-term monitoring's utility. The use of clinical terms like epinephrine without sufficient context for a general ML audience may be inappropriate. Its relevance to long-term monitoring for acute event detection could be more clearly articulated.
*   The cross-dataset generalization experiment's underlying hypothesis and its contribution to validating a "universal grammar" could be explained more deeply.

4.  **Deficiencies in Figures and Data Consistency:**
*   **Figure 1 (Framework):** The description is insufficient. It is unclear whether the GPT component is part of AtomECG's core pre-training framework or a separate model used only for the next-atom prediction downstream task. A clearer, more detailed diagram with explicit data flow and component roles is needed.
*   **Figure 2 & 3 (Next-atom prediction visualizations):**
    *   The labels "Raw" and "Generation" are ambiguous. It is unclear what portion of the signal serves as input for prediction and what is the generated output. The predictive length (e.g., predicting 250 samples after observing 750 samples) should be clearly indicated.

5. **Empirical rigor**
- Patient-level splits & leakage control:
  Given that the PTB-XL dataset already provides predefined folds split at the patient level, using these official folds is the appropriate choice to ensure patient-level independence and prevent data leakage.

- Statistical significance and reproducibility:
  The reported results appear to be based on single-run experiments without confidence intervals or measures of variability. Including confidence intervals (e.g., via bootstrapping or repeated runs with different seeds) would help quantify statistical reliability.

- Reproducibility:
  Clear disclosure of code, data usage, random seeds, hyperparameters, and compute budget would further improve empirical transparency.

**Questions:**

1. **Deepening Motivation and Differentiation:**
*   Could you elaborate on the specific limitations of relying on human-defined waveform segmentation in ECG language processing, and how modern models like ECGBERT and HeartLang address these? What fundamental differences does AtomECG offer, particularly concerning the discovery of sub-waveform morphologies?
*   AtomECG highlights ECG's inherent periodicity and context-dependency. While VQ-VAE and similar discrete representation learning methods focus on local similarity, you argue they struggle with ECG's contextual ambiguity. Could you more concretely explain why Two-Scale Manifold Alignment (TSMA) is intrinsically better suited for ECG's specific challenges compared to generic VQ-VAE approaches? And, could you provide an ablation study that directly compares TSMA against a naive VQ or standard VQ-VAE quantization mechanism to quantitatively demonstrate its necessity and specific advantages for ECG?

2. **Clarity of Two-Scale Manifold Alignment:**
*   To enhance understanding, could you illustrate with more intuitive diagrams or concrete examples of latent space embeddings how TSMA resolves the ambiguity between local similarity and global cardiac context? Specifically, how does it differentiate and assign atoms for healthy, consistent pathological, and irregular pathological ECG patches, and how does the geometry-aware gradient (Householder transformation) adjust for these nuanced distinctions?

3. **Clarification of Data and Experimental Setup:**
*   Regarding Figure 2 and Figure 3, if signal-length (250, 500, 1000, 2000) refers to the number of samples, please clarify the exact data processing steps. For PTB-XL (stated 10s, 500Hz → 5000 samples), how were 1000 or 2000 signal-length segments generated for next-atom prediction?
*   Could you include the performance of from scratch baselines (e.g., ResNet-18) for the multi-label classification tasks in Table 1 to provide a clearer comparison of the benefits of pre-training?
*   In the next-atom prediction tasks using overlapping patches, please detail how the performance evaluation was conducted. Specifically, how is the accuracy of the predicted atom sequence measured against the ground truth atom sequence? Given that MAE and RMSE reflect signal-level reconstruction accuracy, would atom index-level metrics (e.g., accuracy or perplexity of predicted atoms) also be appropriate and informative?

4. **Deepening Interpretability Analysis:**
*   In Table 3, the hyphenated numbers in the Atom usage column, such as 24-696-701-110, are understood to represent atom subsequences. Could you provide a clear explanation of the criteria used to extract these subsequences? This should include details on subsequence length (fixed or variable), frequency, and the methodology for identifying statistically unique subsequences beyond just p<0.05.
*   Please clarify the meaning of 'Proportion of individuals containing the common subsequence' (ratio 5) in Table 3. Does it represent the percentage of individuals within that diagnostic class who exhibit this specific subsequence? Furthermore, could you provide more in-depth clinical interpretations or visualizations for how these identified subsequences reflect the pathophysiology of the associated diseases, similar to how Figure 4 illustrates epinephrine's effect?
*   The discovery of shared atoms (e.g., atom 357 for CD-LBBB/HYP, atom 113 for MI/STTC) indicating pathophysiological links is highly insightful. Could you provide visualizations of how these shared atoms behave in the latent space and how they effectively capture these underlying clinical connections?



5. **Extension of Ablation Studies:**
*   In Table 13, please provide a more detailed interpretation of the Perplexity values across the different quantization configurations (Manifold, Rotation, Residual). Since Perplexity indicates codebook utilization, how do these numbers specifically support the claims that Manifold improves global structure and Rotation enhances local precision? For instance, when all components are applied (✓ ✓ ✓ ✓), the Perplexity is highest (476/324/202/199); does this indicate the best codebook utilization, and how does this relate to the contributions of each individual component compared to lower Perplexity values (e.g., Manifold only 210, Manifold + Rotation 425)?

---

### Official Review · Reviewer_s6ya · 2025-11-01

**Soundness:** 4
**Presentation:** 4
**Contribution:** 4
**Rating:** 6
**Confidence:** 2

**Summary:**

This paper proposes AtomECG, a novel self-supervised learning framework for electrocardiogram (ECG) representation that treats raw signals as discrete sequences composed of morphological “atoms” governed by an implicit grammar. To realize this, the authors introduce Two-Scale Manifold Alignment (TSMA)—a new quantization scheme that jointly optimizes a global “grammar manifold” and local geometry-aware patch-to-atom alignment via Householder transformations. The method is evaluated across multiple diagnostic tasks, interpretability analyses, long-term monitoring scenarios, and cross-dataset generalization settings. Results consistently demonstrate state-of-the-art performance and strong clinical interpretability.

**Strengths:**

The shift from continuous embeddings to interpretable atoms directly addresses a critical gap in clinical adoption of deep ECG models. The paper convincingly shows that specific atoms map to known pathological patterns (e.g., atom 357 linking LBBB and LVH), offering traceable diagnostic rationale. TSMA is a technically sophisticated contribution. The global grammar manifold ensures codebook coherence across cardiac cycles, while the local Householder-based gradient steering enables context-sensitive atom assignment—crucial for distinguishing, e.g., premature beats from normal complexes with similar morphology. Experiments span multi-label/multi-class diagnosis, next-atom prediction (as a proxy for grammar validity), long-term forecasting on hour-scale recordings, and zero/few-shot cross-dataset transfer. The consistent superiority over 8+ baselines—including recent works like HeartLang and ST-MEM—lends strong support to the claims.

**Weaknesses:**

The notion of an “ECG grammar” is compelling but remains somewhat metaphorical. While next-atom prediction validates sequential structure, the paper doesn’t characterize grammatical rules (e.g., transition probabilities, syntactic constraints). Is the grammar Markovian? Hierarchical? More formal modeling would strengthen the claim.
Although VitalDB includes intraoperative artifacts, the paper doesn’t systematically evaluate performance under controlled noise conditions (e.g., baseline wander, EMG interference). Given the fine-grained patching (length=16 @ 500Hz = 32ms), susceptibility to high-frequency noise is plausible.
HeartLang—a strong generative SSL baseline—is included, but recent contrastive methods like CLOCS or Intra-Inter Subject SSL are only briefly cited. A more thorough comparison with the latest eSSL frameworks (e.g., those using lead-augmentation or patient-level invariance) would bolster confidence in SOTA claims.

**Questions:**

The zero-shot transfer from MIMIC/VitalDB to PTB-XL is impressive. Could you clarify whether the codebook atoms themselves generalize (i.e., same indices map to similar morphologies across datasets), or if it’s primarily the grammar manifold structure that transfers?
Have you consulted cardiologists to validate whether the discovered atoms/sequences align with expert intuition beyond post-hoc statistical associations (e.g., via attention maps or prototype visualization)? This would further bridge the ML-clinical gap.
The choice of patch size=16 seems critical. How sensitive is performance to this hyperparameter? Would adaptive patching (e.g., based on instantaneous heart rate) improve robustness across variable cycle lengths?
Are there diagnostic categories where AtomECG underperforms (e.g., subtle ischemia without overt ST changes)?

---

### Note · Authors · 2025-11-13

I have read and agree with the venue's withdrawal policy on behalf of myself and my co-authors.